# Children’s Mental Health: Discrepancy between Child Self-Reporting and Parental Reporting

**DOI:** 10.3390/bs12100401

**Published:** 2022-10-19

**Authors:** Alejandra Caqueo-Urízar, Alfonso Urzúa, Ester Villalonga-Olives, Diego Atencio-Quevedo, Matías Irarrázaval, Jerome Flores, Cristian Ramírez

**Affiliations:** 1Instituto de Alta Investigación, Universidad de Tarapacá, Arica 1000000, Chile; 2Escuela de Psicología, Universidad Católica del Norte, Antofagasta 1270460, Chile; 3Pharmaceutical Health Services Research Department, University of Maryland School of Pharmacy, Baltimore, MD 21201, USA; 4Escuela de Psicología y Filosofía, Universidad de Tarapacá, Arica 1000000, Chile; 5Centro de Justicia Educacional (CJE), Pontificia Universidad Católica de Chile, Santiago 8940855, Chile; 6Departamento de Psiquiatría, Facultad de Medicina, Hospital Clínico Universidad de Chile, Santiago 8380453, Chile; 7Institute for Depression and Personality Research (MIDAP), Santiago 8380453, Chile

**Keywords:** mental health, system of evaluation of children and adolescents (sistema de evaluación de niños y adolescentes; SENA), parent–child discrepancies, multiple informant approach

## Abstract

(1) Background: Discrepancies between children’s self-reports and their parents’ reports on mental health indicators are associated with measurement errors or informant bias. However, they are a valuable tool in understanding the course of child psychopathology. This study aims to determine the level of discrepancies between parents’ perceptions and children’s self-reports in mental health indicators in Northern Chile. (2) Methods: A System of Evaluation of Children and Adolescents self-report (Sistema de Evaluación de Niños y Adolescentes, SENA) was responded to by 408 students between 8 and 13 years old and their parents. (3) Results: Children reported a significantly higher frequency of emotional problems, defiant behavior, and executive functions as compared to their parents’ responses. (4) Conclusions: There is a disjunction between the report of parents and children, which could originate in poor family communication.

## 1. Introduction

Mental health in children and adolescents is a topic that has gained considerable visibility over the last few years [1]. As stated in the Global Burden of Disease of the World Health Organization, it has become one of the main causes of disability in the population between 10 and 24 years old [2]. The concern about mental health indicators at early ages arises from the interaction of multiple factors that can trigger maladaptive reactions in different contexts [3], for which reason a multi-informant assessment approach has been adopted including the reports of those who share close relationships or significant time with the index patient [4]. In the case of the pediatric population, one of the most widely used sources of information, in addition to the child’s own self-report, is that of their parents or caregivers [5].

Although the use of multiple informants in mental health assessment in the infant population benefits the understanding of the psychological functioning of children, discrepancies have been observed between the self-reports and the data provided by parents [6], and also between parents and teachers [7,8], generating new challenges for clinical practice, research, and theory related to child psychiatry and psychopathology [9].

These discrepancies have been extensively studied [10,11], showing that children as young as 6 years old may report independently regarding their health, as compared to parental reporting [12,13]. Furthermore, studies in the 1970s indicated that girls tend to be more reliable informants than boys [14]. Moreover, when externalized problems are analyzed, it seems that parents tend to be more precise than their children; however, when internalized symptoms are analyzed, there seems to be less agreement about which group reports symptoms better [15]. This disagreement stems from the fact that children value their behavior more positively than parents do [10]; although, it becomes more complicated towards adolescence, where they seem to report poorer health than parents, especially in emotional health [16]. 

Various studies support that these discrepancies are based on an underestimation of anxious and depressive symptoms by the parents [17,18,19,20]. In this sense, it has been observed that children tend to report more internalized symptoms such as depression and anxiety than their parents do.

However, when externalizing symptoms such as defiant behavior or hyperactivity are under evaluation, some studies have found a higher level of agreement between both groups [9,21].

In that sense, a meta-analysis carried out by Los Reyes et al. [4] including 341 studies published between 1989 and 2014, observed low-to-moderate correspondence between children’s self-report and parents’ report (mean internalizing: r = 0.25; mean externalizing: r = 0.30; mean overall: r = 0.28).

Additionally, there are certain contextual factors that associate with higher agreement or discrepancy between reports. It has been observed that when there is a higher socioeconomic level, parents tend to underestimate the problems related to their children’s mental health, while in lower socioeconomic levels, the opposite happens [22]. Furthermore, family factors, such as parenting style, lack of communication, and conflict between parents and children, have been associated with higher levels of discrepancy in both reports [23,24,25,26,27,28,29,30,31,32], while family cohesion and parental acceptance have shown fewer discrepancies [9,25].

Traditionally, these observed differences have been interpreted as a function of measurement errors and informant bias [26]. However, such discrepancies may be significant to understand the nature and course of child and adolescent psychopathology, as they may reflect underlying family problems, which potentially contribute to the development of psychopathologies [27]. 

Regarding Chilean culture, the parent–child relationship is significantly framed in a pedagogic–affective relationship of unidirectional transmission, both of values and of interpersonal experiences of recognition and affection, even reaching a parental determinism [28]. This is complemented with a complex relationship of interdependence between children and parents [28,29], which would lead the children to act as “good child” or “without problems” as a support response to the parents, leading to a more limited and biased view of their children’s symptoms [30]. 

Likewise in Chile, there has been little research on the differences between the reports of the respondents [1]. A study by Urzúa et al. [31] addressed the discrepancies and similarities between parents and children, concluding that the different groups were more likely to have different scores, especially in the psychological dimensions. The aim of this study is to determine the level of discrepancies in mental health indicators in parents’ perception and self-reporting of their children in Northern Chile. The hypothesis of the study is the existence of discrepancies between the children’s report versus parents’ perception.

The findings are intended to contribute to the adequate detection of mental health problems in these children and lay the groundwork for the implementation of mental health promotion, prevention, and treatment programs in the child population.

## 2. Materials and Methods

### 2.1. Participants

A convenient sample was used. The initial sample was composed of 5203 elementary and high school students from public, subsidized, and private educational institutions in Arica. For this study, 408 elementary school students were selected, of whom 56.8% (*n* = 232) were girls. In total, 40% (*n* = 163) belonged to the fourth grade, 25% (*n* = 102) to the fifth grade, and 35% (*n* = 143) to the sixth grade. The mean age was 9.97 (SD = 0.96). Sociodemographics can be seen in Table 1.

### 2.2. Instuments

#### Ad-Hoc Socio-Demographic Scale: Used to Identify the Sex, Age, and Grade of Students

The System of Evaluation of Children and Adolescents (Sistema de Evaluación de Niños y Adolescentes, SENA) [32]: The instrument is composed of 9 questionnaires, to assess a broad spectrum of emotional and behavioral problems at three age levels: Infant (3–6 years), Primary (6–12 years), and Secondary (12–18 years). The different questionnaire models have a multi-dimensional approach made up of three blocks of scales: problem scales, vulnerability scales, and personal resources scales.

For this study, the results are reflected in different problem indexes:(a)Global Index of Problems: general summary of the level of discomfort and general affectation presented by people under evaluation. It is calculated from the average of the scales of internalized and externalized problems.(b)Index of Emotional Problems: it summarizes the level of internalized problems presented by the person evaluated, constituting an indicator of emotional alterations and symptoms related to affective or mood disorders. It is composed of the scales of Anxiety (It’s hard for me to make decisions), Depression (It’s hard for me to find things that I really enjoy), Social Anxiety (I get nervous when there are too many people near me), and Somatic Complaints (I wake up tired in morning).(c)Index of Behavioral Problems: indicator of external and disruptive behavioral manifestations that burst into the environment generating interpersonal conflicts and hindering the normal development of activities. It is composed of the scales of Anger Control Problems (When I get angry, I scream to others), Aggression (I make fun of other people for fun), and Defiant Behavior (I do what I want even if I get grounded).(d)Index of Problems in Executive Functions: difficulties in the set of cognitive processes that intervene in goal-directed behavior, allowing planning, goal setting, performance monitoring, and inhibition of ineffective responses. It is composed of the scales of Attention Problems (My teachers say I don’t pay attention in classes), Hyperactivity-Impulsivity (It’s hard for me to stay seated), and Emotional Regulation Problems (It’s hard for me to understand my feelings).

Recently, Sánchez-Sánchez, Fernández-Pinto, Santamaría, Carrasco y del Barrio [33] found that subscales’ reliability was adequate (α > 0.7) in Spain. For this investigation, the versions Self-report—Elementary (134 items) and Family—Elementary (124 items) were used for the parents’ report.

### 2.3. Procedure

This study was approved by the Ethics Committee of the University of Tarapacá and was carried out in compliance with the ethical principles from the Helsinki Declaration for Research in Humans. Informed consent was requested from parents after explaining the purpose and scope of the study, and later the consent was given by the students themselves. Later, at least two trained interviewers were present to answer questions, in the company of the main teacher of the class. The duration of each session was approximately 45 min.

### 2.4. Data Analysis

In the first instance, descriptive analyses of the study variables were carried out. In comparing groups, the condition of normality was tested with the Kolmogorov–Smirnov test, and it was considered that the criteria were met, so the mean comparison analyses were performed using *t*-tests for paired-samples [34]. First, *t*-tests were conducted to compare the averages of children’s and parents’ reports of problem rates. Second, the comparison of means between children’s self-report and their parents’ report of the subscales that composed each index of problems was carried out. Student and parent observations were considered as groups, so parents’ scores were not paired with their children’s.

All analyses were performed using the Statistical Package for the Social Sciences (SPSS) software version 25.0 [35].

## 3. Results

As shown in Table 2, the mean reported by students tends to be higher in all indexes as compared to their parents. The Global Index shows the greatest differences in averages. Moreover, both the Index of Emotional Problems, and the Index of Executive Functions have significant differences. On the contrary, the differences between the scores reported in the Index of Behavioral Problems and Personal Resources do not present significant differences between the student and parents’ groups.

As shown in Table 3, for the Index of Emotional Problems, the variables appear higher in the child’s self-report compared to what is reported by the parents, except for the anxiety scale. The highest statistically significant difference is observed in the somatic complaints. Regarding the Index of Behavioral Problems, the mean difference in the scale of defiant behavior is statistically significant, where the reported symptomatology is higher for the child than for his or her parents. Finally, for the Index of Problems in Executive Functions, the reports from the students are higher than the reports from parents, being that the Index of Problems in Emotional Regulation is the only one that presents a statistically significant difference. 

## 4. Discussion

The aim of this study was to determine the level of discrepancies in mental health indicators between parents’ perceptions and children’s self-reports for students in Northern Chile. The findings of this research show significant differences between children’s self-reporting compared to their parents’ reporting, where children perceive more difficulties on emotional and executive functions compared to their parents.

Regarding emotional problems, it was observed that children report more symptoms than their parents can perceive. This is in line with previous studies that have systematically shown the existence of significant differences between children’s and their parents’ reporting of mental health indicators, particularly in internalized problems [17,18,19,20,36]. The existing literature has hypothesized that, due to the intrapersonal nature of internalized symptomatology, children are more sensitive to it than their parents, making it difficult for parents to correctly assess these difficulties [10]. Nevertheless, this could also imply that due to problems in communication between children and their parents, there are difficulties in detecting their children’s emotional problems.

In terms of problems in executive functions, it is observed that the children reported significantly more difficulties regulating their emotions than their parents did. This may be due to the age range of the sample (9 to 11 years, on average), since this period is characterized by an increased academic demand and body changes, affecting their levels of anxiety and discomfort, so it is to be expected that emotional regulatory skills may be particularly compromised [37].

When analyzing parent–child discrepancies in behavioral problems, no significant differences were observed overall, which is consistent with the previous research [4,9,21,38]. These results imply that the behavioral elements are the most relevant when predicting the level of agreement between parents and children, since these symptoms are usually concrete, observable, severe, and unpleasant, so parents tend to be more willing to report such behaviors.

Finally, it is worth mentioning that this study has limitations. The first is related to the representativeness of the population, both because of the small sample size and the non-probability sampling technique. Secondly, sociodemographic variables such as gender or age were not assessed as a source of discrepancy, while there is evidence that suggests that those factors play a role in the levels of discrepancy between parents and children [39,40]. Finally, the cross-sectional nature of the study did not allow the trajectories of these discrepancies to be determined over time. Future research should be conducted on larger and more diverse samples to improve the generalizability of the results and should consider sociodemographic characteristics of the sample, focusing on understanding the trajectories of parent–child discrepancies over time.

Within the implications of this study, the findings appear to account for a certain “disconnection” between parents and children. Possible reasons for this lack of congruence could be triggered by the prevailing economic system in which both parents work outside the home and in some cases with long working hours, which determines that children spend more time alone, generating fewer instances of time for the family system [41,42]. In the local reality, this is accentuated by the mining activity in the area, since parents move to these mining centers by shift systems that generally imply seven by seven (seven days of work at the mine and seven days of rest at home), which could be negatively impacting the dynamics of the family group [43,44].

This disconnection has also been associated by several studies with adverse effects on children’s mental health indicators, suggesting that poor parental supervision is associated with less behavioral self-control in adolescents [45], as well as less parent–child connection, and leads to lower orientation towards the success of the latter [46]. Finally, the existence of family–work conflicts has been associated as a critical mediating effect on the emergence of maladaptive perfectionist behaviors in children, especially among females [47].

These results suggest the design of interventions that promote greater and better communication within the family, since one of the main causes for consulting for mental health problems in children and adolescents is that their caregivers cannot perceive such difficulties [48], being that this domain is a protective factor in the appearance of mental health problems in children and adolescents [9,25].

## 5. Conclusions

In the studied sample, there are significant differences between children’s self-reporting and their parents’ reporting of mental health indicators, with a tendency for children to perceive greater symptomatology as compared to their parents. These discrepancies could be reduced in the future with the design of interventions that promote communication and optimize interactions in the parent–child relationship.

## Figures and Tables

**Table 1 behavsci-12-00401-t001:** Sociodemographics.

		*n*		Percentage (%)	
Gender	Male	176		43.2	
	Female	232		56.8	
Grade	4th	163		40	
	5th	102		25	
	6th	143		35	
	Min	Max	Mean	Median	SD ^a^
Age	8	13	9.97	10	0.96

Notes: ^a^ SD = Standard Deviation.

**Table 2 behavsci-12-00401-t002:** T-test of Global Indexes reported by children and their parents.

Variable	Report		T	Mean Difference	Sig.
	Children	Parents			
	M (SD) *	M (SD) *			*p*
Global Index of Problems	52.7 (8.09)	39.9 (7.57)	2.474	12.828	0.014
Index of Emotional Problems	54.2 (9.09)	48.9 (8.78)	1.967	3.660	0.050
Index of Behavioral Problems	49.9 (9.68)	46.3 (8.27)	1.333	1.111	0.183
Index of Problems in Executive Functions	52.5 (9.26)	38.9 (8.09)	2.631	2.328	0.009
Index of Personal Resources	42.5 (10.49)	39.2 (8.98)	0.899	0.740	0.369

Note: * M (SD) = Mean (Standard Deviation).

**Table 3 behavsci-12-00401-t003:** *T*-test between children and parents’ report scores.

Variable	Report		T	Mean Difference	Sig.
	Children	Parents			
Index of Emotional Problems	M (SD) *	M (SD) *			*p*
Anxiety	49.1 (9.06)	50.9 (10.84)	−2.560	−1.794	0.011
Depression	56.3 (13.16)	55.3 (11.90)	1.186	1.049	0.236
Social Anxiety	52.6 (9.87)	47.0 (9.40)	8.343	5.642	0.000
Somatic Complains	54.9 (11.03)	48.6 (10.89)	2.316	6.230	0.021
Index of Behavioral Problems	M (SD) *	M (SD) *			*p*
Anger Control Problems	49.8 (10.94)	49.1 (10.91)	0.956	0.737	0.339
Defiant Behavior	51.8 (13.19)	49.9 (10.02)	2.379	1.938	0.018
Aggression	49.8 (10.93)	49.1 (7.46)	1.349	3.578	0.178
Index of Problems in Executive Functions	M (SD) *	M (SD) *			*p*
Attention	52.8 (10.20)	48.2 (9.42)	1.708	4.549	0.088
Hyperactivity	50.4 (10.36)	45.3 (9.47)	1.933	5.139	0.054
Emotional Regulation	53.5 (11.29)	45.1 (10.68)	2.254	8.370	0.024

Note: * M (SD) = Mean (Standard Deviation).

## Data Availability

The data presented in this study are available on request from the corresponding author.

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
