# Peer review of "Children’s Mental Health: Discrepancy between Child Self-Reporting and Parental Reporting"

_behavsci, 2022, doi:10.3390/bs12100401_

Round 1
Reviewer 1 Report
This is an interesting study about the difference in Discrepancy between Child Self-Reporting and Parental Reporting, but it can be improved.
The introduction must be more concise. Also, the authors need to state a clear hypothesis of the study.
In Table 1, please state and Median Age. Also, the demographic data of the parents need to be shown.
In the section Results, the data presented in the Tables must not be repeated. The mean value is not enough; add Standard deviation.
Which test did the authors use to check the distribution of the data?
It is also unclear whether the answers of parents and their children are matched.
Author Response
This is an interesting study about the difference in Discrepancy between Child Self-Reporting and Parental Reporting, but it can be improved.
The introduction must be more concise. Also, the authors need to state a clear hypothesis of the study.
Response: We appreciate the reviewer's comments. We have revised and shortened the introductory section to incorporate the study hypothesis.
In Table 1, please state and Median Age. Also, the demographic data of the parents need to be shown.
Response: As requested, the median age was added in Table 1. The demographic data of the parents was not added since the Family version of the instrument used (SENA) does not ask for such information.
In the section Results, the data presented in the Tables must not be repeated. The mean value is not enough; add Standard deviation.
Response: The redundant information was deleted. The Standard Deviations were added to tables 2 and 3.
Which test did the authors use to check the distribution of the data?
Response: Thanks for your suggestion, the requested information was added in “2.4. Data Analysis” Section of Materials and Methods.
In comparing groups, condition of normality was tested with the Kolmogorov–Smirnov test and it was considered that the criteria were met, so mean comparison analyses were performed using T tests for paired-samples [34]
It is also unclear whether the answers of parents and their children are matched.
Response: The requested information was added in “2.4. Data Analysis” Section of Materials and Methods.
Student and parent observations were considered as groups, so parents' scores were not paired with their children's.
Reviewer 2 Report
I am happy to see papers such as this written. As a clinician who has seen many children with their parents in a psychiatric office, I have seen exactly what you report. Having said this, the children are much more open with the clinician when the parents are absent (out of the room). In lines 96-102 you referred to the Chilean culture and that it often leads to a biased view on the part of the parents toward their children's symptoms. It would have been interesting for you to have added in information regarding whether the parents would see the answers their children gave. I am certain that children worry about the impact their answers will have on their parents. They do not want to hurt their parents, and they are sometimes worried about their parents becoming angry our upset if they find out what answers they gave. I also agree with you, that the socioeconomic background of the families interviewed would have made a more impactful study.
Overall, It was a good study which confirms the existing literature. The information regarding the Chilean miners and their time away from the family unit was also a new observation that could be later explored or compared to other parenting styles.
Author Response
I am happy to see papers such as this written. As a clinician who has seen many children with their parents in a psychiatric office, I have seen exactly what you report. Having said this, the children are much more open with the clinician when the parents are absent (out of the room). In lines 96-102 you referred to the Chilean culture and that it often leads to a biased view on the part of the parents toward their children's symptoms. It would have been interesting for you to have added in information regarding whether the parents would see the answers their children gave. I am certain that children worry about the impact their answers will have on their parents. They do not want to hurt their parents, and they are sometimes worried about their parents becoming angry our upset if they find out what answers they gave. I also agree with you, that the socioeconomic background of the families interviewed would have made a more impactful study.
Overall, It was a good study which confirms the existing literature. The information regarding the Chilean miners and their time away from the family unit was also a new observation that could be later explored or compared to other parenting styles.
Response: We appreciate the comments made by the reviewer. In all my years of experience, this must be one of the first times I have received a clinician's assessment, which is much appreciated since they are ultimately the professionals who work with patients and their families. Regarding whether parents saw their children's answers, unfortunately, even when their son's or daughter's results were made available, a very low number of parents requested those answers. This shows the lack of interest of many families in what is happening with their children, without this comment implying that these families are to blame, Chilean families currently experience high levels of stress and worry about aspects such as work, finances, etc., which do not allow them to have more quality time for parenting. In short, families may lack the time to become more deeply involved in these issues.
Reviewer 3 Report
Congratulations on the research you have done.
The abstract, introduction, methods and discussion are very well and clearly structured.
I think it would be appropriate to redefine the conclusion with an idea of ​​what the direction is for the future and to reduce this discrepancy
Reviewer
Author Response
Congratulations on the research you have done.
The abstract, introduction, methods and discussion are very well and clearly structured.
I think it would be appropriate to redefine the conclusion with an idea of ​​what the direction is for the future and to reduce this discrepancy.
Response: We appreciate positive comments on our work. It is arduous to conduct research when the samples are children and adolescents as well as their parents, so we appreciate the recognition.
Regarding the conclusions, we have included future actions to reduce the discrepancy in the report.
Round 2
Reviewer 1 Report
Thank for accepting my sugestion.
The manuscript is better now, but the extensive
Editing of English is required.
